# QUERY EFFICIENT DECISION BASED SPARSE ATTACKS AGAINST BLACK-BOX DEEP LEARNING MODELS

**Viet Quoc Vo, Ehsan Abbasnejad, Damith C. Ranasinghe**
The University Of Adelaide
{viet.vo,ehsan.abbasnejad,damith.ranasinghe}@adelaide.edu.au

## ABSTRACT

Despite our best efforts, deep learning models remain highly vulnerable to even tiny adversarial perturbations applied to the inputs. The ability to extract information from *solely* the output of a machine learning model to craft adversarial perturbations to black-box models is a *practical* threat against real-world systems, such as Machine Learning as a Service (MLaaS), particularly *sparse attacks*. The realisation of sparse attacks in black-box settings demonstrates that machine learning models are more vulnerable than we believe. Because, these attacks aim to *minimize a number of perturbed pixels*—measured by $l_0$ norm—required to mislead a model by *solely* observing the decision (*the predicted label*) returned to a model query; the so-called *decision-based setting*. But, such an attack leads to an NP-hard optimization problem. We develop an evolution-based algorithm—*SparseEvo*—for the problem and evaluate it against both convolutional deep neural networks and *vision transformers*. Notably, vision transformers are *yet* to be investigated under a decision-based setting. SparseEvo requires significantly fewer queries than the state-of-the-art sparse attack *Pointwise* for both untargeted and targeted attacks. The attack algorithm, although conceptually simple, is competitive with only a limited query budget against the state-of-the-art gradient-based *whitebox* attacks in standard computer vision tasks such as ImageNet. Importantly, the query efficient SparseEvo, along with decision-based attacks, in general, raises new questions regarding the safety of deployed systems and poses new directions to study and understand the robustness of machine learning models.

## 1 INTRODUCTION

In spite of the impressive performance achieved from deep neural network (DNN) models on a variety of vision tasks, a flurry of research on adversarial attacks over the last few years have demonstrated the vulnerability of deep learning models to tiny, maliciously crafted perturbations applied to their inputs (Szegedy et al., 2014). These malicious perturbations, although imperceptible to humans, are able to evade and mislead DNNs. Thus, embedding DNNs in systems creates a new attack surface as well as the incentive for malevolent actors to strike systems such as autonomous cars or machine learning models as a service (MLaaS) employed in real-world applications such as self-driving cars (Chen et al., 2015), Google Cloud Vision or Amazon Rekognition.

In a black-box setting, an adversary may access all or only the *top-1* predicted label and score—*a score-based* setting (Chen et al., 2017)—or simply the predicted label for a given input—*a decision-based* (Brendel et al., 2018) setting. Importantly, the similarity measure, used to quantify the imperceptibility of the perturbation, can describe an attack as a dense attack—$l_2, l_\infty$ norm constrained adversarial attack—or a sparse attack—$l_0$ norm constrained adversarial attack.

Significantly, score-based and decision-based settings present a *practical threat model* for deployed systems; the latter being particularly more threatening to model owners and applications. Because, an adversary is still capable of exploiting the very minimal information exposed—*the top-1 predicted label*—for constructing an perturbation. Importantly, while dense attacks (Athalye et al., 2018; Shukla et al., 2021; Ilyas et al., 2018) are widely explored, *sparse attacks have not drawn much attention*. This potentially leads to a lack of knowledge on model vulnerabilities to this perturbation regime. From a security standpoint, sparse attacks are particularly as threatening as dense

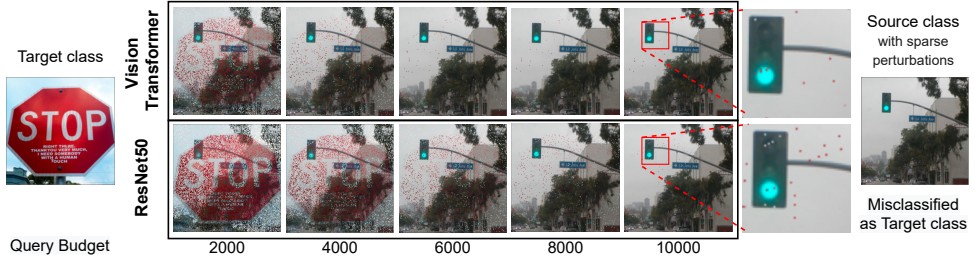

Figure 1: Targeted Attack. Malicious instances generated for a sparse attack with different query budgets using our SparseEvo attack algorithm employed on black-box models built for the `ImageNet` task. With an extremely sparse perturbation *(78 perturbed pixels over a total of 50,176 pixels)*, an image with ground-truth label `traffic light` is misclassified as a `street sign`.

attacks. Therefore, investigating sparse perturbation regimes is as pivotal and necessary as dense perturbation counterparts; in this study, we spend our efforts to extensively investigate the robustness of DNNs against sparse attacks.

**Scope of the Study–Vision Transformers and Convolutional Networks.** Attention-based architectures introduced by Cordonnier et al. (2020); Ramachandran et al. (2019); Touvron et al. (2021), particularly the *Vision Transformer* (ViT) model proposed by Dosovitskiy et al. (2021), can be competitive or even outperform convolution-based architectures (Bhojanapalli et al., 2021; Carion et al., 2020). Existing studies have *not* considered adversarial attacks in $l_0$ norm constraint based perturbation regimes against ViT, although a few studies have explored robustness against $l_2$ and $l_\infty$ norm constraints (Shao et al., 2021). This raises a critical security concern for reliable deployment of real-world applications based on vision transformers. Therefore, our efforts will focus on investigating a method capable of evaluating the robustness of convolutional DNNs as well as transformer networks to understand the fragility of ViT in relation to CNNs under $l_0$ norm adversarial attacks.

**An NP-Hard Problem.** Yielding sparse perturbations is incredibly difficult as minimizing $l_0$ norm leads to an NP-hard problem (Modas & Moosavi-Dezfooli, 2019; Dong et al., 2020). Existing sparse attacks in black-box settings, particularly in decision-based scenarios, have a key shortcoming—the algorithms require a large number of model queries to achieve sparsity and invisibility. Consequently, we propose a novel evolutionary algorithm based sparse attack method in the decision-based setting, we refer to as **SparseEvo**. The method is significantly more query efficient than the state-of-the-art counterpart—Pointwise (Schott et al., 2019). We illustrate an example of a targeted attack with our proposed algorithm in Fig. 1 on the standard computer vision task, `ImageNet`.

**A need for Query Efficiency.** In decision-based or black-box settings, achieving query efficiency with high attack success rate is crucial to adversarial objective. Because: i) adversaries are able to carry out attacks at scale; ii) the cost of mounting the attack is reduced; and iii) adversaries are capable of bypassing a system that can employ methods to recognize malicious activities as a fraud based on pragmatically large number of successive queries with analogous inputs and thwart their attacks. Further, from a defense perspective, the lower number of queries significantly reduces the evaluation time of both trained models and defense mechanisms. Therefore, query efficient attack algorithms facilitate research in designing new defenses, model architectures as well as benefit MLaaS providers by enabling the evaluation of their models prior to deployment at scale.

We summarize our contributions and results below:

- We formulate a novel sparse attack—SparseEvo—an evolution-based algorithm is capable of exploiting access to solely the *top-1 predicted* label from a model to search for an adversarial example in the model's input space whilst minimizing the number of perturbed pixels required to mislead the model.

- Our attack algorithm can significantly reduce the number of model queries compared with the sate-of-the-art counterpart, Pointwise. Further, SparseEvo achieves comparable success to PGD$_0$—the state-of-the-art *white-box* attack—in terms of attack success rate with a limited query budget.

- We conduct the *first* vulnerability evaluation of a Vision Transformer (ViT) on the standard computer vision task `ImageNet` in a decision-based and $l_0$ norm constrained setting. We compare results with ResNet to assess the relative robustness of the ViT model.

## 2 BACKGROUND AND RELATED WORK ON SPARSE ATTACKS

**Adversarial Attack Primer.** Three different criteria can be used to categorize adversarial attacks (Chen et al., 2020); the attack goal, similarity measure used to quantify the imperceptibility of the perturbation and the level of access to information (threat model). In terms of the attack goal, an adversarial attack is either untargeted (an input is simply misclassified) or targeted. Meanwhile, adversarial attacks can be classified into two sub-categories: white-box or black-box according to the threat model. In the white-box setting, an adversary has full knowledge and access to the machine learning model (Goodfellow et al., 2014; Madry et al., 2018; Xu et al., 2019; Carlini & Wagner, 2017) whereas in the black-box setting, solely the outputs of a model are exposed or accessible to the adversary. In the black-box context, an adversary can access all or *top-1* predicted score—a score-based setting (Suya et al., 2020; Chen et al., 2017; Guo et al., 2019)—or simply the predicted labels of a given input—a decision-based (Brendel et al., 2018; Cheng et al., 2020) setting. Imperceptibility, based on a similarity measure, can describe an attack as a dense attack—$l_2, l_\infty$ norm constrained adversarial attacks—or a sparse attack—$l_0$ norm constrained adversarial attacks.

**Sparse Attacks.** The main aim of sparse attacks is to minimize the number of perturbed pixels required to mislead a target machine learning model. Only a handful of works have investigated sparse attacks and these works can be broadly categorised based on various degrees of adversarial access to a model.

**White-box methods.** To realize sparse attacks in a white-box setting, SparseFool attack introduced by Modas & Moosavi-Dezfooli (2019) employed the idea of $l_1$ relaxation from (Andrei & Ion, 2015) and exploited low mean curvature of decision boundaries for $l_0$ minimization. JSMA (Papernot et al., 2017) constructed a saliency map for an input to search for high impact pixels on the model's decision. Recently, Croce & Hein (2019) introduced $PGD_0$ that projects the adversarial perturbation yielded by PGD (Madry et al., 2018) to the $l_0$ ball. This attack method is capable of generating significantly lower $l_0$ perturbation and was shown to outperform other white-box algorithms. Therefore, we use the $PGD_0$ algorithm as *an ideal case baseline* to compare the success achievable in a black-box setting.

**Score-based methods.** (Su et al., 2019) proposed the One-Pixel attack based on a differential evolutionary algorithm. Although the One-Pixel method is capable of searching and obtaining the most sparse perturbation, its attack success rate (ASR) on large neural networks and high resolution images is relatively low. Importantly, the method requires significant number of queries because it modifies one pixel at a time while the input search space, dependent on image resolution, can be enormous. Score-based methods exploit information exposed from a change in confident score to alter a pixel-subset in an input image; a model owner may prevent this leakage by only exposing the top-1 predicted label to a model query.

**Decision-based methods.** In the decision-based setting, only the top-1 predicted label of a DNN model is exposed to adversaries. Now, perturbing an input image slightly will not expose subtle changes in the output corresponding to the perturbation; since only the predicted class label is revealed. Therefore, a decision-based attack is the most restrictive and challenging scenario. Most existing decision-based attack algorithms are *dense attacks (the objective is to minimise $L_2$ or $L_\infty$ distortion). Interestingly, these methods, including BA (Brendel et al., 2018), HSJA (Chen et al., 2020), QEBA (Li et al., 2020), NLBA (Li et al., 2021), PSBA (Zhang et al., 2021), Sign-OPT Cheng et al. (2020) or the covariance matrix adaptation evolution strategy (CMA-ES) based method for face recognition tasks in (Dong et al., 2019), can be adapted to a sparse attack setting by a projection to $L_0$-ball; however this is not effective, as we show later in Appendix A.7. Although CMA-ES (Dong et al., 2019) is an evolutionary algorithm, albeit for a dense attack, the formulation requires individuals of a population to be real number vectors that can be sampled from a Gaussian distribution. Thus, CMA-ES is well suited to the problem of dense attacks. In contrast, the optimization problem in a sparse attack ($L_0$ constrained) aims to minimize the number of perturbed pixels. Importantly, the discrete search space encountered in a sparse attack hinders the adoption of these dense attack algorithms to search for a sparse adversarial example, efficiently.*

To the best of our knowledge, the recent attack—Pointwise (Schott et al., 2019)—applying a greedy search method to find sparse adversarial perturbations is the first decision-based sparse method. This method is effective in untargeted settings and on low resolution datasets, but it is seen to require prohibitively large number of queries to achieve low sparse adversarial perturbations on large scale datasets and in a targeted attack setting (as seen in Section 4). *In summary, the current black-box, sparse adversarial attack approaches still have shortcomings on sparsity and query efficiency. Developing decision-based sparse attacks poses a challenging optimization problem because of: i) limited access to only the decision of a target model; and ii) the NP-hard problem of $l_0$ norm constrained optimization.*

## 3 PROPOSED METHOD

### 3.1 PROBLEM FORMULATION

In our sparse attack setting, we are given a normalized source image $\boldsymbol{x} \in [0,1]^{C \times W \times H}$ and its corresponding ground truth label $y$ from the label set $\mathbb{Y} = \{1, 2, \cdots, K\}$ where $K$ denotes the number of classes, $C$, $W$ and $H$ denotes the number of channels, width and height of an image, respectively. The classifier that we aim to attack is $f : \mathbb{R}^{C \times W \times H} \to \mathbb{Y}$; *our access is limited to its output label*. In a targeted setting, $\boldsymbol{x}$ is perturbed such that the instance $\tilde{\boldsymbol{x}} \in \mathbb{R}^{C \times W \times H}$ obtained is misclassified to a desired class label $\tilde{y} \in \mathbb{Y}$ selected by the adversary. We refer to the desired class of the input $\boldsymbol{x}$ as the *target class* and its ground-truth class as the *source class*. In an untargeted setting, the adversary manipulates input $\boldsymbol{x}$ to change the decision of the classifier to any class label other than its ground-truth, *i.e.* $\tilde{y} \in \mathbb{Y}$ where $\tilde{y} \neq y$. Formally, a sparse adversarial attack (either targetted or untargetted) to find the best adversarial instance $\boldsymbol{x}^*$ can be formulated as a constrained optimization problem:

$$\boldsymbol{x}^* = \arg\min_{\tilde{\boldsymbol{x}}} \|\boldsymbol{x} - \tilde{\boldsymbol{x}}\|_0 \qquad \text{s.t.} \quad f(\boldsymbol{x}^*) = \tilde{y}. \tag{1}$$

where $\|\|_0$ is the $\ell_0$ norm denoting the number of perturbed pixels. The optimization problem in equation 1 aiming to minimize the number of perturbed pixels leads to an NP-hard problem (Modas & Moosavi-Dezfooli, 2019; Dong et al., 2020). Thus, the solution to the optimisation problem is non-trivial given the constraint and the fact that $f$ is not differentiable in this setting.

### 3.2 SPARSEEVO ATTACK ALGORITHM

We devise an efficient parametric search method—SparseEvo—based on an evolutionary algorithm approach to search for a desirable solution through an iterative process of improving upon potential solutions. Through a process of recombination, mutation, fitness evaluation and selection, the quality of a population improves over time to yield a desirable solution. Importantly, our evolution-based search method does not require prior knowledge about the underlying target model, such as model architecture or model parameters to construct a fitness function for assessing potential solutions. Consequently, this method detailed in Algorithm 1 and Fig. 2 is well-suited for solving the non-trivial optimization problem in equation 1 in a black-box setting and provides a possible remedy for the NP-hard problem. We detail formulation of the algorithm in the following.

**Defining a Dimensionality Reduced Search Space.** In applying a parametric search method to the problem, each *candidate solution* can be defined as a *parameter set* consisting of coordinates and RGB values defining all perturbed pixels of an adversarial input in the search space $\mathbb{R}^{C \times W \times H}$. Naively applying a generic parametric search method to seek potential solutions—parameter sets— as observed in One-pixel algorithm (Su et al., 2019), is not effective because the number of queries to the model grows rapidly with respect to the input image size and the number of perturbed pixels.

We propose two techniques to reduce the search space. To facilitate a parametric search method, instead of searching for parameters defining coordinates and RGB values of each perturbed pixel, we propose to solely search for parameters defining coordinates of pixels in the source image to perturb—i.e. image we aim to craft adversarial perturbations for. Constructing all candidate solutions which are parameter sets in a form of coordinate values is dependent on the number of perturbed pixels and hinders the method implementation. Therefore, we vectorize each *candidate solution* in a population as a *binary vector* $\boldsymbol{v} \in \{0,1\}^N$ where 0-bits and 1-bits denotes non-perturbed and perturbed pixels respectively and $N$ is the total number of pixels of an

image. Each element of $\boldsymbol{v}$ corresponds to a pixel and the position $i$ of each element is identified by a mapping function $\phi(n, m)$. Here, we employ a simple flattening technique defined by a mapping function $\phi(n, m) = n + W \times (m - 1)$ where $n$, $m$ are coordinates of a pixel, and $W$ is the width of an image to reduce the search space further. For the color values of these perturbed pixels, we select RGB values from their corresponding pixels in a starting image from the target class (we aim to misclasify the source image to the target class in a targeted attack). We illustrate a source image and a starting image in the context of the algorithm in Figure 2. All candidate solutions—*binary vectors*—can be changed and evolved over iterations until a desirable solution is reached. *Thus our parametric search method essentially transforms to one that will discover the minimum set of most effective pixels to inject into the source image to construct an adversarial example.* Surprisingly, this method is shown to be an extremely effective strategy for a decision-based sparse attack.

The original search space $\mathbb{R}^{C \times W \times H}$ is now transformed to the new search space $\{0, 1\}^N$ where $N = WH$ is the total number of pixels. In other words, a search space on RGB values and $n$, $m$ coordinates is transformed into a search space on $i = \phi(n, m)$ without exploring RGB values. As a result, these techniques lead to a reduction in the size of the search space when compared with the original search space.

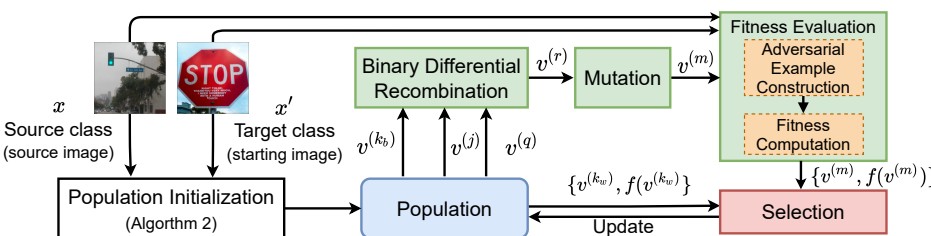

Figure 2: An illustration of SparseEvo algorithm. *Population Initialization* creates the first population generation. This population is evolved over iterations through *Binary Differential Recombination*, *Mutation*, *Fitness Evaluation* (Adversarial Example Construction and Fitness Computation) and *Selection* stages. The source and starting images (used in a targeted attack) are employed to create the initial candidate solutions —binary vector representations—at Population Initialisation and to construct an adversarial example based on a candidate solution $\boldsymbol{v}^{(m)}$ at Fitness Evaluation stage.

---

**Algorithm 1:** SparseEvo

---

**Input:** source image $\boldsymbol{x}$, starting image $\boldsymbol{x}'$, source label $y$, target label $y^*$, model $f$
    population size $p$, initialization rate $\alpha$ mutation rate $\mu$, query limit $T$

1   $t \leftarrow 0$; $\mathbb{V}, \boldsymbol{G} \leftarrow \text{InitialisePopulation}(\boldsymbol{x}, \boldsymbol{x}', f, p, \alpha)$
2   $k_w \leftarrow \arg\max_k(\boldsymbol{G})$, $k_b \leftarrow \arg\min_k(\boldsymbol{G})$    `// Find best and worst individuals`
3   **for** $t = 1, \cdots, T$ **do**
4      Uniformly select $\boldsymbol{v}^{(j)}, \boldsymbol{v}^{(q)} \in \mathbb{V} \setminus \boldsymbol{v}_{k_b}$ at random
5      Yield $\boldsymbol{v}^{(r)}$ using equation 5 and $\boldsymbol{v}^{(k_b)}, \boldsymbol{v}^{(j)}, \boldsymbol{v}^{(q)}$        `// Recombination`
6      Yield $\boldsymbol{v}^{(m)}$ by uniformly altering a fraction $\mu$ of all 1-bits of $\boldsymbol{v}^{(r)}$ at random `// Mutation`
7      Construct $\tilde{\boldsymbol{x}}$ using equation 2, with $\boldsymbol{x}$, $\boldsymbol{x}'$ and $\boldsymbol{v}^{(m)}$
8      Calculate $g(\tilde{\boldsymbol{x}})$ using equation 3 and $f(\tilde{\boldsymbol{x}})$      `// Fitness computation`
9      **if** $g(\tilde{\boldsymbol{x}}_o) < \boldsymbol{G}_{k_w}$ **then**          `// Selection`
10         $\boldsymbol{G}_{k_w} \leftarrow g(\tilde{\boldsymbol{x}})$
11         $\boldsymbol{v}_{k_w} \leftarrow \boldsymbol{v}^{(m)}$
12      $k_w \leftarrow \arg\max_k(\boldsymbol{G})$, $k_b \leftarrow \arg\min_k(\boldsymbol{G})$
13 **end for**
14 Construct $\tilde{\boldsymbol{x}}$ using equation 2 with $\boldsymbol{x}$, $\boldsymbol{x}'$ and $\boldsymbol{v}^{(k_b)}$   `// Build adversarial example`
15 **return** $\tilde{\boldsymbol{x}}$

---

**Fitness Evaluation.** Prior to describing the other phases of the algorithm, we describe the Fitness Evaluation employed for determining the goodness of a candidate solution necessary for the *Population Initialization* and the *Fitness Evaluation* stages, first.

*Adversarial Example Construction.* Since a candidate solution—a binary vector $\boldsymbol{v}$—is used to construct an adversarial example, its fitness is measured by computing an optimization objective for its corresponding adversarial example. Therefore, we first yield an adversarial example corresponding to $\boldsymbol{v}$ based on the following with $c, n, m$ representing a channel and two coordinates of a pixel.

$$\tilde{\boldsymbol{x}}_{c,n,m} \leftarrow (1 - \boldsymbol{v}_i)\boldsymbol{x}_{c,n,m} + \boldsymbol{v}_i\boldsymbol{x}'_{c,n,m} . \tag{2}$$

*The Fitness Function Formulation.* A fitness function should reflect the optimization objective. In the score-based setting, the objective is to optimize loss such that a given input can be misclassified, a reasonable choice for the fitness function is based on output scores as in (Alzantot et al., 2019; Qiu et al., 2021). However, in our problem, the objective to minimize $l_0$ distortion directly results in an NP-hard problem. To alleviate this computational burden, Modas & Moosavi-Dezfooli (2019) relaxed $l_0$ to $l_1$ norm to construct the white-box attack, SparseFool and had access to the output scores, unlike in a decision-based setting. Nonetheless, in the decision-based setting, we find that optimizing $l_2$ norm provides a better alternative than $l_1$. Therefore, in this paper, we formulate our fitness function $g$ (for the targeted attack) as:

$$g(\tilde{\boldsymbol{x}}) \leftarrow \begin{cases} \|\boldsymbol{x} - \tilde{\boldsymbol{x}}\|_2, & \text{if} f(\tilde{\boldsymbol{x}}) = \tilde{y} \\ \infty, & \text{otherwise} \end{cases} , \tag{3}$$

Where $\tilde{\boldsymbol{x}}$ is an image constructed using equation 2 and $\tilde{y}$ is a target class. A similar fitness function for the untargeted attack can be formulated as equation 3 but the constraint is now $f(\tilde{\boldsymbol{x}}) \neq y$.

**Population Initialization.** Recall, our search objective is to discover a minimum perturbation represented by a binary vector—*candidate solution*. Hence, we initialize a population of $p$ different candidate solutions from an *initialized vector* $\boldsymbol{v}^{(o)}$ formulated as following with C number of channels.

$$\boldsymbol{v}_i^{(o)} \leftarrow \begin{cases} 0, & \text{if } \boldsymbol{x}_{c,n,m} = \boldsymbol{x}'_{c,n,m} \ \forall c \in \{1, \cdots, C\} \\ 1, & \text{otherwise} \end{cases} \tag{4}$$

Every candidate is generated by only randomly altering $d$ 1-bits of $\boldsymbol{v}^{(o)}$, where $d = \lfloor \alpha W H \rfloor$, $\alpha$ is an initialization rate. A candidate solution is successfully added to the population, if its fitness score is not $\infty$; we explain our fitness function in equation 3. Otherwise, another $d$ 1-bits is randomly flipped to generate another candidate solution. This process is repeated until all $p$ successful candidates are found and stored in a population set $\mathbb{V}$. The corresponding fitness score of each candidate solution is stored in a fitness score matrix $\boldsymbol{G}$. The pseudocode of Population Initialization phase is detailed in Algorithm 2 in Appendix A.1.

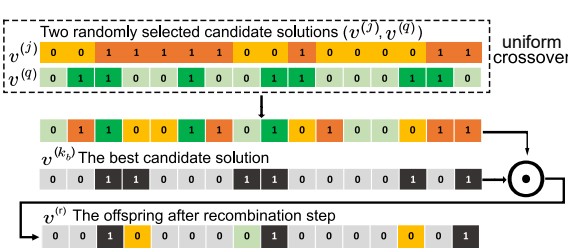

Figure 3: The Binary Differential Recombination is shown in Algorithm 1 (line 6) and equation 5. $\odot$ is an element-wise product, $\boldsymbol{v}^{(k_b)}, \boldsymbol{v}^{(j)}, \boldsymbol{v}^{(q)}$ are the best and two randomly selected candidate solutions from a population respectively.

**Binary Differential Recombination.** In some recombination methods used in genetic algorithms (GA) e.g. k-point or uniform crossover, a couple of parents are mated to produce an *offspring* for the next generation. However, after Population Initialization stage, all first-generation parents are slightly different from each other since all of them are generated from an initialized vector $\boldsymbol{v}^{(o)}$. Consequently, these crossover variants lead to sub-par solutions and low query efficiency. To address this problem, we increase diversity in a population. Inspired from the differential evolutionary (DE) algorithms (Storn & Price, 1997), we create the next generation by mutating and combining multiple existing parents. Nonetheless, applying DE naively is impractical since the mutation operation of DE algorithm adds the weighted difference of multiple selected parents to another parent to yield an offspring. These individuals are vectors in real coordinate space so the offspring can benefit from weighted real-valued difference but it cannot be gained in our proposed search space in which all candidate solutions are binary vectors. Therefore, we propose Binary Differential Recombination scheme—a hybrid method based on the uniform crossover in GA and the notion of mutation in DE.

There are different mutation schemes which can influence the overall performance (Manolis & Vagelis, 2020). In the problem of decision-based attacks, through our empirical results shown in Appendix A.3, we observe that the approach of recombining the best and two selected candidate solutions outperforms others. Hence, we first select two candidate solutions $v^{(j)}, v^{(q)}$ uniformly at random from the population. We then employ *uniform crossover* for selecting each bit from either selected candidate solutions with equal probability to yield a new candidate solution. Subsequently, the best individual $v^{(k_b)}$ in the population is recombined with the new candidate solution by altering all 1-bits of $v^{(k_b)}$ whose corresponding bits in the new candidate solution are 0-bits. Formally, the Binary Differential Recombination can be formulated as:

$$v^{(r)} \leftarrow v^{(k_b)} \odot \texttt{UniformCrossover}(v^{(j)}, v^{(q)}) \tag{5}$$

where $\odot$ is an element-wise product. This operation is visualized in Fig. 3. As a consequence of gaining from the difference between individuals, our method is capable of boosting evolutionary progress as shown in Section 4.

**Mutation.** Diversity in the population is a key factor that enables exploration in the search space to obtain better individuals. As a result, mutation operation aiming to promote this population diversity is a crucial component of our method and every offspring after recombination step can be subject to mutation. In practice, we uniformly select a fraction $\mu$ of all 1-bits of the offspring $v_o$ at random and set these bits to zero. We do not select 0-bits for altering because it hinders the optimization progress and requires more iteration to search for the optimum.

**Selection.** Our simple intuition is that individuals with better fitness values should lead to survival over future generations. In problem 1, a smaller fitness value is better and represents a more imperceptible adversarial example. To this end, if the worst individual in the population has higher fitness value than the offspring's, it will be discarded and the new offspring is then chosen to take its place.

## 4    EXPERIMENTS AND EVALUATIONS

### 4.1    EXPERIMENT SETTINGS

**Attacks and Datasets.** For a comprehensive evaluation of the effectiveness of SparseEvo, we employ two standard computer vision tasks with different dimensions: CIFAR10 (Krizhevsky et al.) and ImageNet (Deng et al., 2009). We compare with the state-of-the-art sparse attack algorithm in Pointwise (Schott et al., 2019) and use the white-box sparse attack $PGD_0$ (Croce & Hein, 2019) to benchmark against the black-box decision-based counterparts. For the evaluation sets, we select a balanced sample set. We randomly draw 1,000 and 200 *correctly* classified test images from CIFAR10 and ImageNet, respectively. These selected images are evenly distributed among the 10 (CIFAR10) and 200 randomly selected (ImageNet) classes. In the *targeted* setting, while each image from CIFAR10 is attacked to flip its ground-truth label to 9 target classes, a set of five target classes are randomly selected for each image from ImageNet to reduce the computational burden of the evaluation tasks. All of the parameter settings are summarized in Appendix A.2

**Models.** For convolution-based models, we use a state-of-the-art architecture—ResNet—(He et al., 2016), particularly, ResNet18 for CIFAR10, achieving 95.28% test accuracy, and a pre-trained ResNet-50 provided by torchvision (Marcel & Rodriguez, 2010) for ImageNet with a 76.15% Top-1 label test accuracy. For attention-based models, we selected a pre-trained ViT-B/16 model obtaining 77.91% Top-1 label test accuracy (Dosovitskiy et al., 2021). Notably, this model was trained by Google on the large scale and high resolution ImageNet dataset.

**Evaluation Measures.** To evaluate the performance of methods, we define a normalised *sparsity measure* as $l_0$-norm distortion divided by the total number of pixels of an image and then compute the *median* of sparsity over an evaluation set—since it is not sensitive to outliers. A measure used to evaluate the robustness of a model is *Attack Success Rate* (ASR). A generated perturbation is successful if it can yield an adversarial example with a sparsity *below a given sparsity threshold*, then ASR is defined as *the number of successful attacks over the entire evaluation set*. In black-box settings, ASR can be calculated at different sparsity thresholds after the assessment of the evaluation set with a given query budget. Notably, there is no query constraint for $PGD_0$. We run $PGD_0$ with different perturbation budgets and ASR is calculated based on the best achieved results.

**Attack initialization (targeted and untargeted).** We need a starting image $x'$ to initialize an attack. For *targeted attacks*, we consider a randomly chosen correctly classified image from the dataset. For *untargeted attacks*, we may perturb the source image by adding a *uniform*, *Gaussian* (Cheng et al., 2020; Chen et al., 2020) or *salt and pepper* noise (Schott et al., 2019) until it is misclassified. In practice, we observe that employing salt and pepper noise for our untargeted attack is more effective.

**Experimental Regime Summary.** We conduct: 1) Attacks against conventional CNNs on `CIFAR10` and `ImageNet` tasks but we defer the results of `CIFAR10` to Appendix A.5; 2) Attacks Against a Vision Transformer on the `ImageNet` task; 3) Compare the robustness of the ViT model with the CNN model; and we defer the following experiments to the Appendix: 4) Attacks against defended models (Appendix A.4); 5) Comparing with an improved PointWise algorithm as a baseline (Appendix A.7); 6) Comparing with dense attacks adapted to a sparse setting (Appendix A.8);. Further, in addition to Figure 1, we provide more visual comparisons in Figure 10.

## 4.2 ATTACKS AGAINST CONVOLUTIONAL DEEP NEURAL NETWORKS

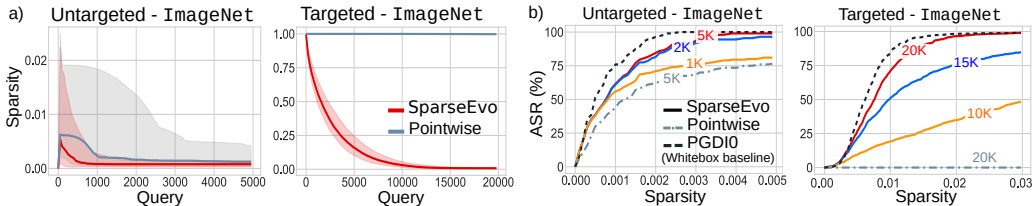

Figure 4: Evaluation set from `ImageNet` using the ResNet50 model with image size (W×H): 224×224. a) Median sparsity with the first and third quartiles used as lower and upper error bars versus the number of model queries; and b) attack success rate versus sparsity thresholds.

**Query Efficiency Evaluation.** Fig. 4a shows the median sparsity against model query budgets on the `ImageNet` task. Our attack consistently outperforms the the Pointwise method in terms of queries and sparsity. In the untargeted setting, SparseEvo achieves a lower sparsity than the Pointwise attack under various query budgets. In the targeted setting, our attack is able to craft adversarial images with extremely sparse perturbation within 20,000 queries for most images from `ImageNet` but Pointwise does not perform well in this task.

**Attack Success Rate.** Fig. 4b illustrates ASR against different sparsity threshold at different query budgets for SparseEvo on the `ImageNet` task and we compare with the best achievement of $PGD_0$ (ideal, whitebox attack) and Pointwise (decision-based sparse attack). In the untargeted setting, we observe that SparseEvo achieves a higher ASR than Pointwise employing a 5,000 query budget with the small budget of 1,000 queries. In the targeted setting, our attack with a 10,000 query budget demonstrates significantly better ASR than Pointwise employing 20,000 queries. Interestingly, a small query budget of 5,000 queries is adequate to achieve the same ASR as the white-box setting in the $PGD_0$ attack in the untargeted setting, while around 20,000 queries achieves comparable performance to the ideal white-box setting for a targeted attack. This is significant for decision-based attacks since adversaries are given very limited access to a the model.

The evaluation results with `CIFAR10` in Appendix A.5 confirm the the observations on the `ImageNet` task and demonstrate the generalizability of the SparseEvo algorithm. We also summarise results at query budgets and attack settings on the two vision tasks in Table 2 in the Appendix.

## 4.3 ATTACKS AGAINST A VISION TRANSFORMER

**Query Efficiency Evaluation.** Fig. 5a shows the median sparsity against the queries. With a limited number of queries, SparseEvo is able to achieve significantly lower sparsity than Pointwise in both targeted and untargeted setting. While our attack is able to converge to a extremely high sparsity after 3,000 and 15,000 queries for untargeted and targeted setting, respectively. Pointwise fails to converge to lower values in both settings.

**Attack Success Rate.** Fig. 5b illustrates that with only 1000 queries, SparseEvo outperforms Pointwise with a 5,000 query budget across all different sparsity thresholds. Notably, in the untargeted

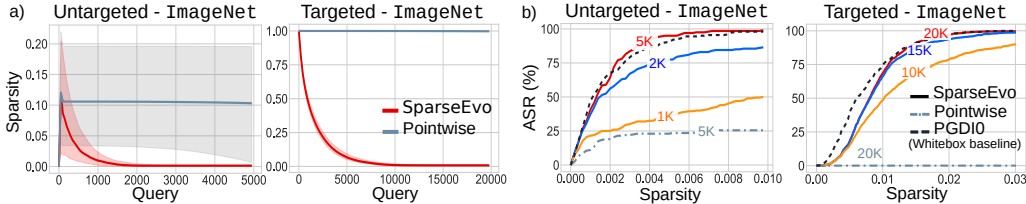

Figure 5: Evaluation set from `ImageNet` using the ViT model with image size (W×H): 224×224. a) Median sparsity with the first and third quartiles used as lower and upper error bars versus the number of model queries; and b) attack success rate versus sparsity thresholds.

setting, SparseEvo with a query budget of 5,000 is able to achieve slightly higher ASR than the ideal white-box $PGD_0$ from a sparsity threshold of 0.002. In the harder, targeted setting—SparseEvo with only 15,000 queries is able to obtain marginally lower ASR than $PGD_0$ whereas with a 20,000 query budget, our attack is as robust as $PGD_0$ when sparsity threshold is larger than 0.01.

## 4.4 COMPARE THE ROBUSTNESS OF THE TRANSFORMER AND THE CNN

In this section, we compare the robustness of ViT and ResNet50 models to sparse perturbation in untargeted and targeted settings. Fig. 6 reports the accuracy of these models over adversarial examples of an evaluation set of 100 images from `ImageNet`. We summarise results at query budgets and attack settings in Table 3 in the Appendix. Overall, we find that the performance of ViT degrades as expected but it appears to be less susceptible than the ResNet50 model. Particularly, in the untargeted setting, the accuracy of ViT across different sparsity thresholds is higher than the ResNet50 model under both SparseEvo and $PGD_0$. Interestingly, SparseEvo only needs a *small query budget of 2,000* to degrade the accuracy of ResNet50 that is similar to white-box $PGD_0$, while up to 5,000 queries are needed to make SparseEvo attack on ViT worse than $PGD_0$. In the targeted scenario, we observe that at a low query budget e.g. 10,000, ResNet50 is much more robust than ViT under SparseEvo whereas at 20,000 queries, the accuracy of both ResNet50 and ViT models is almost analogous and drops to approximately zero when sparse perturbation is larger than 0.02. Notably, SparseEvo with a sufficient query limit e.g. 20,000 is able to maintain its attack effectiveness against both ViT and ResNet50 while attack effectiveness of $PGD_0$ is reduced—demonstrated by lower accuracy scores—when attacking ViT.

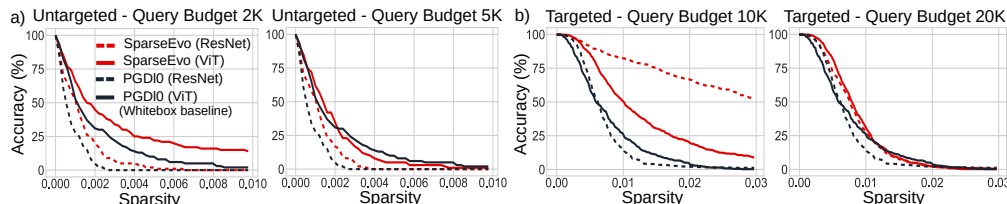

Figure 6: Attack success rate versus sparsity thresholds at different query budgets for the evaluation set from `ImageNet` with ViT vs ResNet. $PGD_0$ is a white-box attack (ideal).

## 5 CONCLUSION

In this work, we propose a new algorithm for a sparse attack—SparseEvo—under a decision-based scenario. Our comprehensive results demonstrate SparseEvo outperforms the state-of-the-art black-box attack in terms of sparsity and ASR within a given query budget. More importantly, in a high resolution and large scale dataset, SparseEvo illustrates significant query-efficiency and remarkably lower sparsity when compared with the existing sparse attacks in the black-box setting. Most notably, our black-box attack, under small query budgets, achieves comparable success to the state-of-the-art white-box attack—$PGD_0$ (for further insights we refer the reader to Appendix A.9).

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

# A APPENDIX

## A.1 POPULATION INITIALIZATION

Algorithm 2 presents pseudo-code for our Population Initialization approach as presented in Section 3.2.

---

**Algorithm 2:** InitialisePopulation

---

**Input:** source image $x$, starting image $x'$, source label $y$, target label $y^*$, model $f$
        population size $p$, initialization rate $\alpha$

1   $\mathbb{V} \leftarrow \emptyset, G \leftarrow \infty$
2   $n \leftarrow \lfloor \alpha W H \rfloor$                  // W, H are image width and height
3
4   Generate a binary vector $v$ using equation 4
5   **for** $t = 1, 2, \cdots, p$ **do**
6      **while** $True$ **do**
7         Generate $v^{(o)}$ by uniformly altering $n$ of all 1-bits of $v$ at random
8         Construct $\tilde{x}$ using equation 2 with $x$, $x'$ and $v^{(o)}$
9         Calculate $g(\tilde{x})$ using equation 3 and $f(\tilde{x})$      // Calculate Fitness Score
10
11         **if** $g(\tilde{x}) < G_t$ **then**
12            $G_t \leftarrow g(\tilde{x})$
13            $\mathbb{V} \leftarrow \mathbb{V} \cup \{v^{(o)}\}$
14            **break**
15      **end while**
16   **end for**
17   **return** $\mathbb{V}, G$

---

## A.2 HYPER-PARAMETERS

We list in in Table 1 the key hyper-parameters used for SparseEvo on the two different evaluation sets across `CIFAR10` and `ImageNet`. This hyperparameter set can be applicable for attacking against ViT-B/16 on large scale and high resolution dataset—`ImageNet`. Notably, we only needed to adjust the mutation rate for when moving from the high resolution to the low resolution `CIFAR10` task; thus, our method provides a robust algorithm that can be easily adopted for different vision tasks.

The image size used in all our `ImageNet` experimental tasks (including experiments on ResNet50 and ViT models) is (3 channels) $\times$ 224 (W) $\times$ 224 (H). This is the standard input size for the pre-trained model (PyTorch) on the `ImageNet` dataset we used.

Table 1: Hyper-parameters setting in our experiments

| Parameters | CIFAR10 | | ImageNet | |
|---|---|---|---|---|
| | Untargeted | Targeted | Untargeted | Targeted |
| **Population size** $(p)$ | 10 | 10 | 10 | 10 |
| **Initialization rate** $(\alpha)$ | 0.004 | 0.004 | 0.004 | 0.004 |
| **Mutation rate** $(\mu)$ | 0.04 | 0.01 | 0.004 | 0.001 |

## A.3 ROBUSTNESS TO HYPER-PARAMETERS AND INVESTIGATING RECOMBINATION AND MUTATION SCHEMES

In this section, we conduct comprehensive experiments to study the impacts of hyper-parameters used in our algorithm and different recombination and mutation schemes we considered. These experiments are carried on 1,000 randomly selected images from `CIFAR10` in an untargeted setting. For the hyper-parameter study, we tune population size or mutation rate at a time while using the scheme of recombining the best and two randomly selected candidates from the population as well as the scheme of mutating only 1-bit binary values.

Fig. 7a shows that with different population sizes and a mutation rate of 0.04, even a small population size of 10 is adequate for SparseEvo to converges rapidly. Our method with a larger population size almost converges to as low sparsity as the population size of 10 after 200 queries. So population size has a small impact on the overall performance of SparseEvo. A mutation rate at 0.04 and fixed population size of 10, the algorithm performs well and converges fastest to a low sparsity compared to others mutation rates as shown in Fig. 7b. Consequently, our attack method is more influenced by the mutation rate but this is not unexpected.

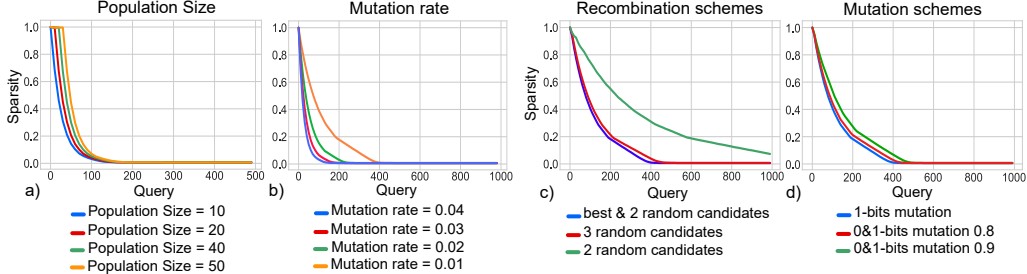

Figure 7: Sparsity versus number of model queries on `CIFAR10` with ResNet18 to show the impacts of different hyper-parameters on SparseEvo.

To evaluate how different schemes of recombination and mutation steps affect our method, we use the population size of 10 and mutation rate of 0.01 and change the recombination or mutation scheme, one at a time. Figure 7c illustrates that recombining three randomly selected individuals does not achieve as high query-efficiency as the scheme of recombining the best and other two

randomly selected from the population. For mutation schemes, we intend to mutate merely 1-bits—*1-bits mutation*—or both 0-bits and 1-bits—*0 & 1-bits mutation*—of a binary vector at a time. For 1-bits mutation scheme, we randomly alter a factor $\mu$ of all 1-bits of a selected binary vector. For schemes mutating both 0-bits and 1-bits, we randomly flipped $n$ 1-bits and $\frac{n(1-\beta)}{\beta}$ 0-bits where $n = \mu\beta$. We find that the scheme of mutating only 1-bits performs marginally better than other schemes with $\beta = 0.8$ and $\beta = 0.9$ because mutating both 0 and 1-bits possible slows down the convergent speed as illustrated in Figure 7d.

### A.4 ROBUSTNESS OF SPARSE ATTACKS AGAINST AN ADVERSARIALLY TRAINED MODEL

In this section, we study the robustness of different sparse attacks against adversarially trained ResNet-18 network on the `CIFAR10` task using $l_\infty$ perturbations—one of the most effective defense mechanisms against adversarial attacks (Athalye et al., 2018). The accuracy of this adversarially trained network is 83.87%. We choose $PGD_0$ (Croce & Hein, 2019), a state-of-the-art *white-box attack* as a baseline for comparison. The adversarial training based models used in this experiment is trained with Projected Gradient Descent (PGD) adversarial training proposed by Madry et al. (2018).

The experiment is conducted on a balance evaluation set withdrawn from `CIFAR10` randomly (we describe the dataset in Section 4.1. Median sparsity against the number of queries is shown in Figure 8. The results indicate that SparseEvo converges faster than the Pointwise attack. Figure 8 also shows the attack success rate (ASR) at different distortion levels and query limits for different attack methods against the adversarially trained model. We observe that our attacks are able to obtain a comparable performance with the ideal white-box $PGD_0$ baseline attacks with a very limited query budget of merely 500 queries. Meanwhile SparseEvo is comparable with Pointwise with a given query budget of 200, and outperforms Pointwise with a query budget of 500.

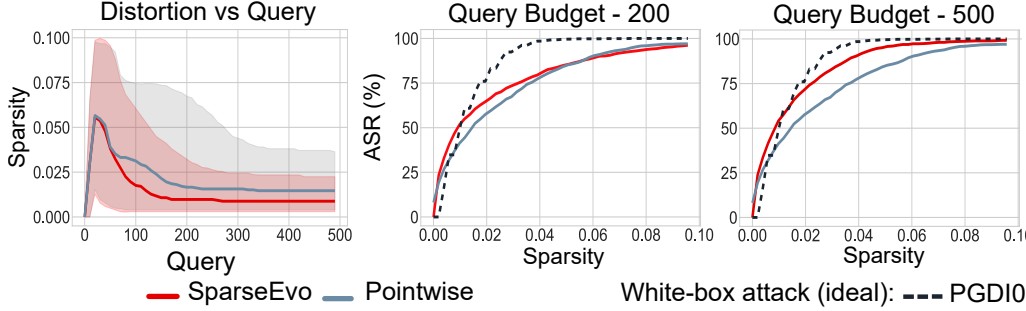

Figure 8: Different sparse attacks against an adversarially trained model on the `CIFAR10` task. We show sparsity versus queries and ASR versus sparsity two different query budgets: 200 and 500.

### A.5 ATTACKS AGAINST A CNN MODEL FOR THE CIFAR10 TASK

Fig. 9a shows the median sparsity against the queries as well as the first and third quartiles used as lower and upper error bars. The figure provides a comprehensive comparison for different attacks on the evaluation set from `CIFAR10` in both untargeted and targeted settings. Our attack consistently outperforms the the Pointwise attack in terms of queries and sparsity. Particularly, in the untargeted setting, our attack is able to craft adversarial images by perturbing extremely low number of pixels, on average within 2,000 queries for most images on `CIFAR10`; while Pointwise only obtains a sparsity of 0.75 for this evaluation set. In the targeted setting, SparseEvo converges to a lower sparsity than the Pointwise attack with a given query budget.

**Attack Success Rate.** Figure 9b illustrates ASR against different sparsity threshold at different query budgets for SparseEvo on the evaluation set from `CIFAR10` and also compare with the best achievement of $PGD_0$ (ideal, white-box baseline) and Pointwise (state-of-the-art black-box sparse attack). In the untargeted setting, we observe that SparseEvo using 200 queries or more achieve higher success rates than Pointwise using 500 queries. Notably, the our black-box sparse attack can

achieve comparable ASR to $PGD_0$ with a small query budget of 500 queries. In the targeted setting, with only 500 queries our attack demonstrates significantly better ASR than Pointwise across all sparsity thresholds, while SparseEvo achieves marginally lower ASR than $PGD_0$ (ideal, white-box baseline) with a query budget of 2,000.

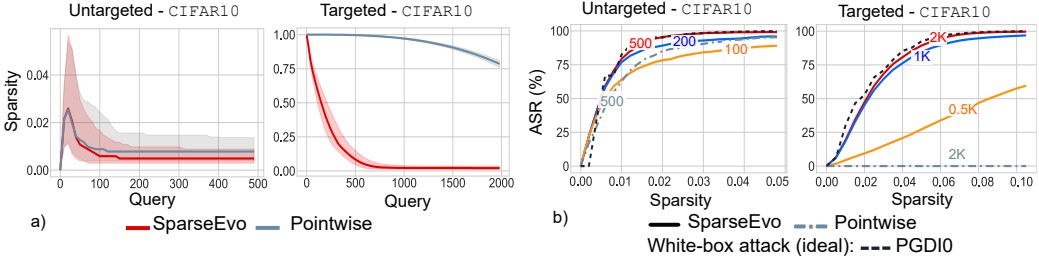

Figure 9: Evaluation set from `CIFAR10` using a ResNet18 model. a) Median sparsity with the first and third quartiles used as lower and upper error bars versus number of model queries; and b) attack success rate (ASR) versus sparsity thresholds.

Table 2: Median sparsity and ASR at different query budgets. A comprehensive comparison among different attacks ($PGD_0$, Pointwise and SparseEvo) on small and large scale balance datasets.

| Setting | Query budget | Methods | CIFAR10 Median | CIFAR10 ASR | Query budget | ImageNet Median | ImageNet ASR |
|---|---|---|---|---|---|---|---|
| Untargeted | | $PGD_0$ | 0.0059 | **99.8**% | | **0.0005** | **100**% |
| | 200 | Pointwise | 0.0078 | 88.0% | 2000 | 0.0016 | 68.0% |
| | | SparseEvo | **0.0049** | 96.5% | | 0.0008 | 96.5% |
| | 500 | Pointwise | 0.0078 | 96.2% | 5000 | 0.0012 | 77.0% |
| | | SparseEvo | **0.0049** | 99.2% | | 0.0008 | 99.0% |
| Targeted | | $PGD_0$ | 0.0703 | **99.8**% | | **0.0061** | 99.0% |
| | 1000 | Pointwise | 0.9612 | 0.0% | 10000 | 0.9997 | 0.0% |
| | | SparseEvo | 0.0311 | 96.5% | | 0.0511 | 48.5% |
| | 2000 | Pointwise | 0.7863 | 0.0% | 20000 | 0.9975 | 0.0% |
| | | SparseEvo | **0.0251** | 99.6% | | 0.0076 | **99.1**% |

Table 3: Accuracy of ResNet50 and ViT under attacks at different query budgets and sparsity thresholds. A comprehensive comparison among different attacks ($PGD_0$ and SparseEvo) on small and large scale balanced evaluation sets from `ImageNet`

| Setting | Methods | Query Budget | ResNet50 | | ViT | |
|---|---|---|---|---|---|---|
| Sparsity | | | 0.002 | 0.004 | 0.002 | 0.004 |
| Untargeted | $PGD_0$ | na | **5**% | **0.0**% | **31**% | 14% |
| | SparseEvo | 2000 | 20% | 5% | 45% | 25% |
| | | 5000 | 17% | **0.0**% | 35% | **7**% |
| Sparsity | | | 0.02 | 0.03 | 0.02 | 0.03 |
| Targeted | $PGD_0$ | na | **2.0**% | 1.2% | 4.4% | **0.2**% |
| | SparseEvo | 10000 | 66.8% | 52.8% | 20% | 9.0% |
| | | 20000 | 2.2% | **0.6**% | **2.4**% | **0.2**% |

## A.6 ALGORITHMIC COMPARISON WITH POINTWISE

In this section, we discuss why SparseEvo is capable of searching for a desirable solution (an adversarial example with a smaller number of perturbed pixels) with much fewer queries.

1. *Greedy vs. Evolutionary approach.* Pointwise chooses to greedily minimize the number of perturbed pixels by randomly selecting and altering one dimension (i.e. single colour channel) of a randomly selected pixel position i,j of the starting image $x' \in R^{C \times W \times H}$ at a time (i.e per query). If the alternation successfully fools the model, it will be retained; otherwise, the change will be discarded. In contrast, SparseEvo evaluates candidate proposals to alter several pixels at a time and all dimensions of a pixel simultaneously to yield new candidates solutions or offspring for the next evolution; so it is able to converge faster and with less queries.

2. *Smaller search space.* Pointwise formulation leads to a search space with a size of $C \times W \times H$ where C is the three RGB channels, W is image width and H is image height. We reduce this search space to $W \times H$ because SparseEvo solely searches for pixel positions but does not try to search for different colors for each pixel (see "Defining a Dimensionality Reduced Search Space" in Section 3.2 and Appendix A.7).

3. *Better scalability to large image sizes.* Given that PointWise only changes one dimension at a time (i.e a pixel), to reduce the number of starting image (target class) pixel values different from the source image (to minimize $l_0$), the random selection method needs to select: i) the same pixel position $i$, $j$; and ii) a different colour channel for the same pixel position $i$, $j$ in subsequent iterations to move a given pixel value $i$, $j$ in a starting image (target class image) to be the same as the source image. While this is more likely in a small image task (with smaller W and H values) like `CIFAR10`, it is far less likely, even within the 20,000 query budget used with large input images in the `ImageNet` task where mean sparsity values for the 1000 test image pairs remain nearly 1.

4. *Iterative improvements to "good" solutions.* Importantly, our approach formulates a search for a solution with the minimum number of perturbed pixels through an iterative process of improving upon good solutions from previous iterations informed by our objective function. In contrast, Pointwise employs a purely random method to select the pixel dimension and position $i$, $j$ to alter.

## A.7 COMPARISON WITH AN IMPROVED POINTWISE ALGORITHM AS A BASELINE

Table 4: Mean sparsity measure at different queries (lower is better) for a targeted attack setting. A comparison between SparseEvo and improved Pointwise on a set of 100 image pairs on `ImageNet` (here PW-$n_p$ denotes PointWise with number of selections set to $n_p$ and italicised fonts indicate the best results for PW.)

| Query Budgets | 1 | 500 | 1000 | 2000 | 4000 | 8000 | 12000 | 16000 | 20000 |
|---|---|---|---|---|---|---|---|---|---|
| PW(published version) | 1.00 | 1.00 | 1.00 | 1.00 | 1.00 | 1.00 | 1.00 | 1.00 | 1.00 |
| PW-4 | 1.00 | 1.00 | 1.00 | 1.00 | 1.00 | 0.99 | 0.97 | 0.93 | 0.88 |
| PW-8 | 1.00 | *1.00* | *1.00* | *1.00* | *0.99* | *0.94* | *0.81* | *0.58* | *0.35* |
| PW-16 | 1.00 | 1.00 | 1.00 | 0.99 | 0.94 | 0.64 | 0.45 | 0.42 | 0.40 |
| PW-32 | 1.00 | 1.00 | 0.99 | 0.95 | 0.71 | 0.54 | 0.50 | 0.46 | 0.42 |
| PW-64 | 1.00 | 1.00 | 0.95 | 0.78 | 0.67 | 0.62 | 0.56 | 0.51 | 0.46 |
| PW-128 | 1.00 | 0.96 | 0.84 | 0.77 | 0.74 | 0.67 | 0.61 | 0.56 | 0.52 |
| SparseEvo | 1.00 | **0.76** | **0.63** | **0.46** | **0.26** | **0.08** | **0.03** | **0.01** | **0.01** |

PointWise randomly selects and alters one dimension (a colour channel) of a randomly selected pixel position $i$, $j$ of an image $x' \in R^{C \times W \times H}$ at a time (i.e per query). Therefore, the Pointwise formulation leads to a search space with a size of $C \times W \times H$ where C is the three RGB channels, $W$ is image width and $H$ is image height. Consequently, it is not scalable to large image sizes, for example ImageNet with a size of $224 \times 224$; this can be observed in Fig. 4 and 5.

In this section, we attempted to make PointWise more query efficient on `ImageNet` by modifying PointWise to perform multiple selections at a time (i.e per query) and perform a series of experiments using different selection parameters $n_p$. Table 4 shows the mean sparsity obtained by our improved Pointwise method with different selection parameter values; $n_p$ = 4, 8, 16, 32, 64, 128. The results

show that the *best performance* of the modified Pointwise algorithm—*PW-8*—is much better than the original implementation but it is still far behind our method. SparseEvo still outperforms our improved Pointwise algorithms across various query budgets.

## A.8 COMPARISON WITH DENSE ATTACKS ADAPTED TO CONSTRUCT SPARSE ATTACKS

We are motivated to investigate if decision-based dense attacks ($l_2$ and $l_\infty$ constrained) such as BA (Brendel et al., 2018), HSJA (Chen et al., 2020), QEBA (Li et al., 2020), NLBA (Li et al., 2021), PSBA (Zhang et al., 2021), SignOPT (Cheng et al., 2020) or RayS (Chen & Gu, 2020) can be adapted to a sparse setting by a projection to $l_0$-ball. This idea is promising because PGD can be successfully adapted to a sparse setting to provide a sparse attack algorithm in a white-box setting. In this section, we conduct a study to evaluate this idea by modifying the HSJA method because it is shown to be a query-efficient decision-based dense attack ($l_2$ and $l_\infty$ constraint), to an $l_0$ constraint algorithm called $l_0$-HSJA. Notably, the same could be done for other methods e.g. QEBA, NLBA, PSBA, SignOPT, or RayS.

Importantly, the authors of HSJA proposed two different ways of gradient estimation purposely formulated for $l_2$ and $l_\infty$ scenarios. However, the $l_0$ distance metric is non-differentiable and therefore is ill-suited for standard gradient descent (Carlini & Wagner, 2017; Fan et al., 2020) so we leverage $l_2$ to estimate the gradient. The difference between the $l_0$-HSJA algorithm and published HSJA is the projection step. Instead of performing $l_2$ and $l_\infty$ projection steps as in HSJA, $l_0$-HSJA performs an $l_0$ projection as in the PGDl0 method. To search for the minimum number of pixels to perturb, we adopt a binary search to minimise $l_0$. At each iteration (with the discovered adversarial sample from HSJA), we perform the following projection procedure:

1. $l_0$-HSJA sorts pixel differences between the sample adversarial crafted by HSJA and the source image.

2. $l_0$-HSJA then performs a binary search for k denoting the minimum number of (perturbed) pixels to retain from the sample adversarial crafted by HSJA. Here, k=$\frac{ur+lr}{2}$ where $lr$ and $ur$ are lower and upper ranges, initialized with 0 and $N$, respectively. $N$ is the total number of pixels in an image.

3. Subsequently, we create a candidate sparse adversarial example by keeping only the top-$k$ pixels of the HSJA crafted adversarial sample and replacing the rest of the pixels of the crafted sample with their corresponding pixel in the source image we plan to fool. These top-$k$ pixels have the least difference to their corresponding pixels. This yields the projected image $x_p$ for evaluation. If the projected sample can mislead a victim model successfully, $ur$ is updated with $k$ (to search for a lower number of perturbed pixels). Otherwise, $lr$ is updated with $k$.

4. This step is repeated until the $ur$ and $lr$ difference is less than or equal to the threshold 1.

For the following iteration of $l_0$-HSJA, we use the projected image $x_p$ to craft a new adversarial example $x_p'$ to attempt to improve upon the projected adversarial example from the current iteration.

The results we obtained, shown in Table 5, illustrate the average sparsity for a set of 100 image pairs on CIFAR10. Our evaluations show that applying $l_0$ projection to dense attacks (formulated for $l_2$ and $l_\infty$ methods) does not yield a query efficient sparse attack aiming to minimize the number of perturbed pixels. We can understand this result, because, at each projection step, the modified $l_0$-HSJA algorithm still requires a large number of queries to determine a projection that minimises $l_0$ (in other words, to determine the minimum number of pixels to retain where the crafted sample is still adversarial).

To the best of our knowledge, there is no efficient method in a black-box decision-based setting to determine how many pixels and which pixels can be selected to be projected such that the perturbed image does not cross the unknown decision boundary of the DNN model. Additionally, the problem of minimizing the number of selected pixels to be projected leads to an NP-hard problem (Modas & Moosavi-Dezfooli, 2019; Dong et al., 2020). Although we use the projected image with the minimum number of perturbed pixels, $l_2$ and $l_\infty$ decision-based attacks require perturbing a whole image in the following iteration, thus the next iteration does not necessarily move the input towards the objective of minimizing the number of perturbed pixels. Thus, $l_0$-HSJA and other dense methods do not provide an efficient algorithm for sparse attacks.

Table 5: Mean sparsity measure at different queries (lower is better) for a targeted setting. A comparison between $l_0$-HSJA and SparseEvo on a set of 100 image pairs on `CIFAR10`

| Queries | 1 | 500 | 1000 | 2000 | 4000 | 8000 | 12000 | 16000 | 20000 |
|---|---|---|---|---|---|---|---|---|---|
| $l_0$-HSJA | 1.00 | 0.82 | 0.95 | 0.92 | 0.92 | 0.95 | 0.95 | 0.94 | 0.94 |
| SparseEvo | **1.00** | **0.36** | **0.027** | **0.025** | **0.025** | **0.025** | **0.025** | **0.025** | **0.025** |

## A.9 A Discussion on Results with the Whitebox Baseline

Notably, $PGD_0$ is an adapted-to-$l_0$version of the PGD attack with a projection. $PGD_0$ simply projects the adversarial example generated by PGD attack onto the $l_0$-ball (we described the process in Appendix A.8 earlier regarding adopting non-sparse decision based attacks). This projection does not guarantee that a projected solution yields the best gradient descent direction for the following iteration of PGD to find an adversarial example that minimises $l_0$. Hence, even with full access to the model, $PGD_0$ may not always yield the optimal solution but rather an approximation. So $PGD_0$ may not always be an upper bound for the attack performance, particularly in the untargeted setting on `ImageNet` as shown in Figure 5(b) and the second plot of Figure 6.

## A.10 Visualization of Sparse Adversarial Examples

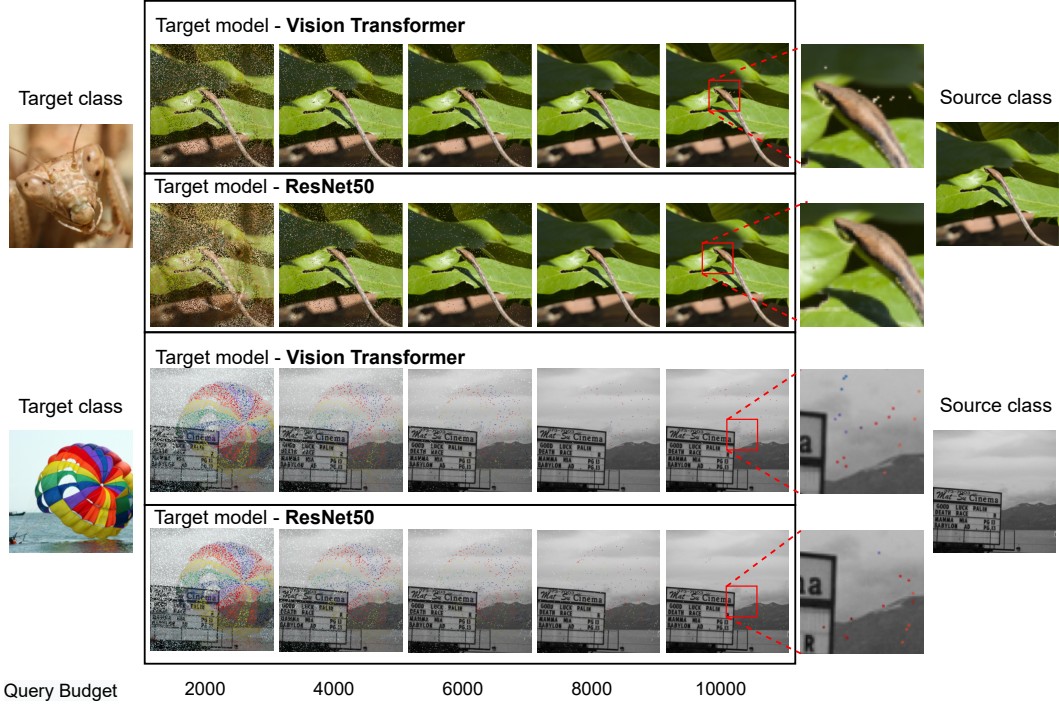

Figure 10: Visualisations from a targeted attack Settings. Malicious instances generated for a sparse attack with different query budgets using our SparseEvo attack algorithm employed on black-box models built for the `ImageNet` task.

