# OpenReview forum: "QUERY EFFICIENT DECISION BASED SPARSE ATTACKS AGAINST BLACK-BOX DEEP LEARNING MODELS"
_ICLR.cc/2022/Conference — ICLR 2022 Poster_

### Official Review · Reviewer_HSPi · 2021-10-31

**Correctness:** 3
**Technical Novelty And Significance:** 1
**Empirical Novelty And Significance:** 2
**Recommendation:** 5
**Confidence:** 3

**Main Review:**

Strengths:
The paper shows promising experimental results for the method. As a paper proposing a blackbox method, it also shows comparison with a white-box attack to showcase its superiority (my concerns are described in the weakness part below).

Weaknesses:
1. The experiment section can be more comprehensive. The submission only compares with one paper on decision-based sparse attack, and that work only shows experiments on MNIST dataset, not ImageNet. (a) The comparison on ImageNet shown in this submission is not fair (the PointWise method sparsity is always 1, which means it basically fails to create any sparsity): if the comparison were to be made, the submission can instead make some minimal adjustments to the baseline method to make it not completely useless. (b) There are many other decision-based attacks on ImageNet. Although most of them are showing results in L-2 metrics (e.g., BA, HSJA, QEBA, NLBA, PSBA, SignOPT, etc.) and some of them show L-\infty metrics (e.g., RayS), many of them can be easily adapted to L-0 case with projections based on my experience. The submission can try to compare with these stronger baselines on ImageNet to showcase its method performance.
2. The paper can discuss its relationship/difference with the existing literature more clearly. For example, using evolutionary methods for decision-based attacks is not an invention by this submission: the paper “Efficient Decision-based Black-box Adversarial Attacks on Face Recognition” has proposed one in 2019. Also, as mentioned above, though many existing decision-based attack papers did not show results on L-0 metrics, they can be adapted easily, thus very related with this paper. The paper should consider a more detailed discussion on its related works and justify its novelty in terms of the proposed method.

Questions:
On the second plot in Figure 6, for the two solid curves, the red curve (SpaEvo (ViT)) even goes lower than the black curve (PGDl0 (ViT)) both at the beginning and the end. Is there an explanation for this observation? Are you using different images for different curves so that the white-box attack PGD is not the upper bound of the attack performance in this plot? Or is the PGD not optimized properly? The “Untargeted-ImageNet” plot in Figure 5(b) is also weird in a similar sense.


**Summary Of The Paper:**

The paper proposes an evolution-based algorithm to conduct a sparse attack against convolutional deep neural networks and vision transformers. The evaluation results show that the proposed method requires fewer model queries than the state-of-the-art sparse attack Pointwise for both untargeted and targeted attacks.

**Summary Of The Review:**

The paper shows good experimental results, but there are some concerns about the experimental part (whether it's fair and valid). Also, the novelty of the method and the relationship with the literature are not discussed in detail.

---

> ### Author Response · Authors · 2021-11-19
> **Response to the Question on the Plot in Figure 6 and Figure 5(b)**
>
> Thank you for your insightful comments. In all our experiments, we used the same image pairs for all attack methods.
>
> For PGDl0 [6], we used the PyTorch implementation provided by the authors. Notably, PGDl0 proposed in [6] is an adapted-to-$L_0$version of the PGD attack with a projection. PGDl0 simply projects the adversarial example generated by PGD attack onto the $L_0$-ball (we described the process in response to your questions earlier regarding adopting non-sparse decision based attacks). This projection does not guarantee that a projected solution yields the best gradient descent direction for the following iteration of PGD to find an adversarial example that minimises $L_0$. Hence, even with full access to the model, PGDl0 may not always yield the optimal solution but rather an approximation at best. *So PGDl0 may not always be an upper bound for the attack performance, particularly in the untargeted setting on ImageNet as shown in Figure 5b and the second plot of Figure 6*.
>
> - [6] F. Croce and M. Hein. Sparse and imperceivable adversarial attacks. International Conference on Computer Vision (ICCV), 2019.
>
> __Actions (please see Rebuttal Revision)__
>
> - We added text to explain the results further in Appendix A.9 because of the limited space in the main body of the paper due to other changes and the expanded related work in *Section 2* to include _*dense*_ decision based attack methods.

---

> ### Author Response · Authors · 2021-11-19
> **2. Discuss relationship/difference with non-sparse ($L_2$ and $L_\infty$ constrained) decision-based attacks**
>
> Thank you for your valuable advice; in the paper, we limited our discussion to sparse attacks but according to your suggestion, we will add non-sparse decision-based attacks, including a discussion of [RefA]. We hope the clarifications below in the context of the study in [RefA] will be helpful till then.
>
> The study in [RefA], in contrast to a sparse attack, demonstrates a dense attack where the objective is to minimise $L_2$ or $L_ \infty$, as done in BA, HSJA, QEBA, NLBA, PSBA, SignOPT. The attack proposed in [RefA] uses covariance matrix adaptation evolution strategy (CMA-ES) and is a great demonstration of an evolutionary algorithm formulation for a dense attack. Although CMA-ES is an evolutionary algorithm, albeit for a dense attack, the formulation requires individuals of a population to be real number vectors that can be sampled from a Gaussian distribution. Thus, CMA-ES is well suited to the problem of dense attacks. In contrast, the optimization problem in a sparse attack ($L_0$ constrained) aims to minimize the _number of perturbed pixels_ that is discrete. This is an NP-hard problem (please see “An NP-Hard Problem” in Section 1 and [4, 5]) where the careful formulation of the evolutionary algorithm we propose is a highly effective and query-efficient solution.
>
> Importantly, the discrete search space encountered in a sparse attack hinders the adoption of these dense attack algorithms to search for a sparse adversarial example, efficiently. Consequently, we first formulate a dimensionality reduced search space and model the solution space as a binary space where possible solutions (individuals of a population) are binary vectors. This is very different from most current evolution methods.
>
> In our paper, to reduce the search space, we presented two simple but efficient techniques (a vectorization and flattening strategy, please kindly see “Defining a Dimensionality Reduced Search Space” in Section 3.2). Additionally, we also proposed the __Binary Differential Recombination__ scheme—a hybrid method based on the uniform crossover in Genetic Algorithms and the notion of mutation in Differential Evolution; please kindly see “Binary Differential Recombination” in Section 3.2).
>
> - [RefA] Yinpeng Dong, Hang Su, Baoyuan Wu, Zhifeng Li, Wei Liu, Tong Zhang, Jun Zhu, “Efficient Decision-based Black-box Adversarial Attacks on Face Recognition”, CVPR 2019
>
> __Actions (please see Rebuttal Revision)__
>
> - We incorporated a non-sparse attacks discussion (including adding [RefA]) (please see page 3, Section 2).
> - We added __*new comparison results*__  from an adaptation of a non-sparse attack to construct a sparse attack algorithm in Appendix A.8 and mentioned them on page 8, Section 4.1: *Experimental  Regime  Summary*
>
> __Reproduced from the paper for convenience__
>
> - [4] A. Modas and P. Moosavi-Dezfooli, S. Frossard. SparseFool: a few pixels make a big difference. Computer Vision and Pattern Recognition (CVPR), 2019
> - [5] X. Dong, D. Chen, J. Bao, C. Qin, L. Yuan, W. Zhang, N. Yu, and D. Chen. GreedyFool: DistortionAware Sparse Adversarial Attack. Neural Information Processing Systems (NeurIPS), 2020.

---

> ### Author Response · Authors · 2021-11-19
> **1(b) Build and compare with an adaptation of the algorithms published for $non-sparse~attacks$ (those for minimizing $l_2$ and $L_\infty$)**
>
> We agree; there are a number of decision-based __dense attacks__ ($L_2$ and $L_\infty$ constrained) such as BA, HSJA, QEBA, NLBA, PSBA, SignOPT, or RayS. For a comparison, modifying these published algorithms for a sparse attack ($L_0$ constraint) is an interesting idea. According to your suggestion, we modified HSJA, which is a query-efficient decision-based dense attack  ($L_2$ and $L_\infty$ constraint), to an $L_0$ constraint algorithm we call $L_0$-HSJA. Notably, the same could be done for other methods.
>
> Importantly, the authors of HSJA proposed two different ways of gradient estimation; purposely formulated for $L_2$ and $L_\infty$ settings. However, the $L_0$ distance metric is non-differentiable and therefore is ill-suited for standard gradient descent [1, 2] so we leverage $L_2$ to estimate the gradient. The difference between the $L_0$-HSJA algorithm and published HSJA is the projection step. Instead of performing $L_2$ and $L_\infty$ projection steps as in HSJA, $L_0$-HSJA performs an $L_0$ projection as in the PGDl0 method. To search for the minimum number of pixels to perturb, we adopt a binary search to minimise $L_0$. At each iteration (with the discovered adversarial sample from HSJA), we perform the following projection procedure in $L_0$-HSJA :
>
> 1. Sorts pixel differences between the sample adversarial crafted by HSJA and the source image.
>
> 2. Perform a binary search for k denoting the minimum number of (perturbed) pixels to retain from the sample adversarial crafted by HSJA. Here, k=$\frac{ur + lr}{2}$ where $lr$ and $ur$ lower and upper ranges, initialized with 0 and N, respectively. N is the total number of pixels of an image.
>
> 3. Then create a candidate sparse adversarial example by keeping only the top-k pixels of the HSJA crafted adversarial sample and replacing the rest of the pixels of the crafted sample with their corresponding pixel in the source image we plan to fool. These top-k pixels have the least difference to their corresponding pixels. This yields the projected image $x_p$ for evaluation. If the projected sample can mislead a victim model successfully, $ur$ is updated with k (to search for a lower number of perturbed pixels). Otherwise, $lr$ is updated with k.
>
> 4. This step is repeated until the $ur$ and $lr$ difference is $\leq$ the threshold 1.
>
> For the following iteration of $L_0$-HSJA, we use the projected image $x_p$ to craft a new adversarial example $x'_p$  to attempt to improve upon the projected adversarial example from the current iteration.
>
> Table 2: _Mean sparsity measure at different queries (lower is better) for a targeted setting. A comparison between $L_0$-HSJA and SparseEvo on a set of 100 image pairs on CIFAR-10._
>
> |Queries|1|500|1000|2000|4000|8000|12000|16000|20000|
> |---|---|---|---|---|---|---|---|---|---|
> |$L_0$-HSJA|1.00|0.82|0.95|0.92|0.92|0.95|0.95|0.94|0.94|
> |**SparseEvo**|__1.00__|**0.42**|**0.19**|**0.08**|**0.04**|**0.03**|**0.03**|**0.03**|**0.03**|
>
> The results  shown in Table 2, illustrate the average sparsity for a set of 100 image pairs on CIFAR-10. Our evaluations show that applying $L_0$ projection to dense attacks (formulated for $L_2$ and $L_\infty$ methods) does not yield a query efficient sparse attack aiming to minimize the number of perturbed pixels. We can understand this result, because, at each projection step, the modified $L_0$-HSJA algorithm still requires a large number of queries to determine a projection that minimises $L_0$ (in other words, to determine the minimum number of pixels to retain where the crafted sample is still adversarial).
>
> To the best of our knowledge, there is no efficient method in a black-box decision-based setting to determine how many pixels and which pixels can be selected to be projected such that the perturbed image does not cross the unknown decision boundary of the DNN model. Additionally, the problem of minimizing the number of selected pixels to be projected leads to an NP-hard problem [3, 4]. Although we use the projected image with the minimum number of perturbed pixels, $L_2$ and $L_\infty$ decision-based attacks require perturbing a whole image in the following iteration, thus the next iteration does not necessarily move the input towards the objective of minimizing the number of perturbed pixels. Thus, L0-HSJA and other dense methods do not provide an efficient solution.
>
> __Actions__
>
> - We added this discussion, __*new results*__, references to decision-based __dense attacks__ ($L_2$ and $L_\infty$ constrained) to our revision (see pg. 3, Section 2 & pg. 8 & Appendix A.8).
>
> __Reproduced from our paper for convenience__
>
> - [1] Nicholas Carlini et al. “Towards Evaluating the Robustness of Neural Networks”, SP2017
> - [2] Yanbo Fan et al., “Sparse Adversarial Attack via Perturbation Factorization”, ECCV2020
> - [3]. A. Modas et al. Sparsefool: a few pixels make a big difference. CVPR, 2019.
> - [4] X. Dong et al. GreedyFool: Distortion- Aware Sparse Adversarial Attack. NeurIPS, 2020.

---

> ### Author Response · Authors · 2021-11-19
> **1(a) Improve the published PointWise ($sparse~ attack$, i.e. minimise $l_0$) algorithm and compare as a baseline.**
>
> Thank you for your insightful comments. Notably, PointWise is the first decision-based sparse attack, ours, to the best of our knowledge, is the second such attack algorithm, thus we are limited by what we can compare against.
>
> We have done our best to follow your suggestions to take the published PointWise algorithm and to improve it. We discuss and share our findings below.
>
> We attempted to make PointWise more query efficient. We improved the published Pointwise algorithm. PointWise randomly selects and alters one dimension (a colour channel) of a randomly selected pixel position i,j of an image $x' \in R^{C\times W \times H}$ at a time (i.e per query). We modified PointWise to perform multiple selections at a time (i.e per query) and perform a series of experiments using different selection parameters $n_p$.
>
> __Table 1__ shows the mean sparsity obtained by our improved Pointwise method with different selection parameter values; $n_p$ = 4, 8, 16, 32, 64, 128. The results show that the performance of the modified Pointwise algorithm is much better than the original implementation but it is still far behind our method. SparseEvo still outperforms our improved Pointwise algorithms across various query budgets.
>
> Table 1: _Mean sparsity measure at different queries (lower is better) for a targeted attack setting. A comparison between SparseEvo and improved Pointwise on a set of 100 image pairs on ImageNet (here PW-$n_p$ denotes PointWise with number of selections set to $n_p$)_
>
> |  Queries    | 1 | 500 | 1000 | 2000 | 4000 | 8000 | 12000 | 16000 | 20000 |
> |--------------|---|-------|--------|---------|--------|---------|---------|----------|-----------|
> |  PW(published version) | 1.00    | 1.00   | 1.00    | 1.00    | 1.00    | 1.00    |1.00    | 1.00    | 1.00     |
> |  PW-4      | 1.00 | 1.00  | 1.00    | 1.00    | 1.00    | 0.99    | 0.97    | 0.93    | 0.88     |
> |  PW-8 ($n_p$ value for best PW result)     | 1.00 | 1.00  | 1.00    | 1.00    | 0.99    | 0.94    | 0.81    | 0.58    | 0.35     |
> |  PW-16    | 1.00 | 1.00  | 1.00    | 0.99    | 0.94    | 0.64    | 0.45    | 0.42    | 0.40     |
> |  PW-32    | 1.00 | 1.00  | 0.99    | 0.95    | 0.71    | 0.54    | 0.50    | 0.46    | 0.42     |
> |  PW-64    | 1.00 | 1.00  | 0.95    | 0.78    | 0.67    | 0.62    | 0.56    | 0.51    | 0.46     |
> |  PW-128  | 1.00 | 0.96  | 0.84    | 0.77    | 0.74    | 0.67    | 0.61    | 0.56    | 0.52     |
> |  **SparseEvo**  | **1.00** | **0.76**  | **0.63**   | **0.46**    | **0.26**    | **0.08**    | **0.03**    | **0.01**    | **0.01**     |
>
>
> __Actions (please see Rebuttal Revision)__
> - We added the __*new results*__ and discussion to the revised manuscript (see Appendix A.7)

---

### Official Review · Reviewer_8jjY · 2021-11-02

**Correctness:** 3
**Technical Novelty And Significance:** 3
**Empirical Novelty And Significance:** 3
**Recommendation:** 6
**Confidence:** 3

**Main Review:**

Pros:

1. The experimental performance is really good in terms of it being a decision-based sparse attack.
2. Using the L2/L1 distance as the fitness function and using an evolution algorithm instead of some estimated gradients to generate adversarial examples is novel.

Cons:

1. The comparison with the pointwise attack in the targeted attack experiments is somehow unfair. SparseEvo relies on a random target image to generate the adversary while the pointwise attack doesn't. It would be better to find a way to let the pointwise attack leverage the target image or adapt another black-box attack for doing the sparse attacks.
2. I am wondering what's the image size used in the ImageNet experiments? Since you only reduce the search space by a factor of the # channels (typically 3). So I am wondering how the scalability of SparseEvo is against the big images.

**Summary Of The Paper:**

This paper proposes a black-box decision-based spare attack based on the evolution algorithm (called SparseEvo).  The authors test their method on two types of classification models and two popular vision datasets: ResNet (CIFAR10 and ImageNet) and Vision Transformer (ImageNet).  Through the comparison with Pointwise attack (for efficiency and sparsity) and PGD0 (for success rate), SparseEvo achieves good performance in both success rate and efficiency.

**Summary Of The Review:**

This paper proposes a novel black-box decision-based space adversarial attack method based on the evolution algorithm. The basic idea is to use the L2/L1 distance with the original image as the fitness function to adjust the current images towards the target images. The experimental results are good. I am only concerned a little bit about the comparison in the targeted attack since it is somehow unfair (see in the Main Review).

---

> ### Author Response · Authors · 2021-11-19
> **2. What's the image size used in the ImageNet experiments? And clarifications on the scalability of SparseEvo.**
>
> Thank you for point this out; sorry we missed mentioning this. The image size used in all our ImageNet experiments (including experiments on ResNet50 and ViT models) is (3 channels) x 224 (W) x 224 (H). This is the standard input size for the pre-trained model (PyTorch) on the ImageNet dataset we used.
>
> In our approach, it is possible to achieve a reduction by a factor that is more than the number of channels. Let us try to clarify how SparseEvo is able to reduce the search space by a larger factor than 3. The search space is enormous to begin with: W x H x R x G x B where R, G, B are each 256 different values (on an integer scale) and W is image width and H is image height. However, SparseEvo strategy is to solely search for the best pixel positions (please kindly see “Defining a Dimensionality Reduced Search Space” in Section 3.2) because we do not try to search for different colors for each pixel. _Therefore, SparseEvo can reduce the search space to (W x H) of an image and significantly more scalable than PointWise_.
>
> __Actions (please see revised version)__
>
> - We updated the caption in Fig. 4 with the image size of ImageNet
> - We updated the caption in Fig. 5 with the image size of ImageNet
> - We also updated Appendix A.2 to state that we used the standard image size for ImageNet.

---

> ### Author Response · Authors · 2021-11-19
> **1. Clarifications on the comparison with PointWise targeted attack and the suggestion to adapt another black-box attack**
>
> Thank you for your constructive comments.
>
> We are sorry if our description was not clear on how we run and collect results for the Pointwise attack. So let us explain further there.
>
> * __In a targeted attack Setting__. Pointwise and SparseEvo methods need a sample of a starting image from the target class and a source image from the source class for which we wish to find a sparse perturbation (see _Fig. 1_). In our evaluation of a targeted attack, to be fair, we __*selected exactly the same pairs of source and starting images for both SparseEvo and Pointwise methods*__. So we used the same target image (or the starting image) as our attack method for Pointwise. So, Pointwise does leverage the target image.
>
> * __In an untargeted attack setting__. You are right, both Pointwise and SpaEvo do not need a target image (what we call a starting image) to initialize an attack as the source image simply needs to be misclassified to any other class. Importantly, in the untargeted setting, to be fair, we (SpaEvo) adopted the same initialization method used by PointWise.
>
> Figure 2 may be confusing since we show an illustration in the targeted attack setting. We will improve the caption in _Fig. 2_ to make it clearer that this is for a targeted attack and that in an untargeted setting, to be fair, we adopted the same initialization method used by PointWise.
>
> We hope this clarifies the concern raised by the Reviewer.
>
> We also thank you for your suggestion to adapt another black-box attack, we have attempted to employ projection-based methods employed in the PGDl0 white-box sparse attack method in decision-based dense (non-sparse) attacks. We provide our reformulations of these published works with results in response to __Reviewer HSPi__, to avoid redundancy, kindly see the response there. _Essentially, our results demonstrate that SpaEvo remains significantly more effective and query efficient_.
>
> __Actions (please see Rebuttal Revision)__
>
> - We updated the caption in Fig. 2
> - We updated *Attack Initialization* in Section 4.1 to further clarify the different initializations for targeted and untargeted attacks.
> - We adopted a dense attack and added the __*new results*__ to Appendix A.8

---

### Official Review · Reviewer_DgSu · 2021-11-03

**Correctness:** 4
**Technical Novelty And Significance:** 3
**Empirical Novelty And Significance:** 3
**Recommendation:** 6
**Confidence:** 2

**Main Review:**

- The proposed methods are well-motivated and novel. The paper is easy to follow for an adequately prepared reader. Prior work is sufficiently discussed.
- The experiments are convincing and the experiment results show the effectiveness of the proposed attack.
- The amount of detail is good, it seems sufficient to reproduce results.

**Summary Of The Paper:**

This work proposes a novel sparse attack method called SparseEvo. Based on evolution algorithm, the SparseEvo searchs a sparse adversarial perturbation in limited query budget. It can significantly reduce the queries compared with the SOTA method, i.e., Pointwise. The paper also conduct the first vulnerability evaluation of a ViT on ImageNet in a decision-based and $l_0$ norm constrained setting.

**Summary Of The Review:**

Overall, I think this paper is a good one.

---

> ### Author Response · Authors · 2021-11-19
> **Authors' response to Reviewer DgSu**
>
> Thank you for encouraging comments and appreciating our efforts.
>
> We will release __all source code and project artifacts__ in the final version of the paper to ensure reproducibility as well as support future research in the field. If needed, we are happy to provide the source code and artifacts to the reviewers in an anonymized repository.
>
> __Actions (please see Rebuttal Revision)__
>
> - Project source code and artifacts will be open sourced here: *https://sparseevoattack.github.io/* (we ensured anonymity)

---

### Official Review · Reviewer_t1fV · 2021-11-05

**Correctness:** 4
**Technical Novelty And Significance:** 3
**Empirical Novelty And Significance:** 3
**Recommendation:** 6
**Confidence:** 5

**Main Review:**

***Strengths***
- The paper is largely clear and well-written.
- The experimental results are solid, and experiments are carried out on vision datasets and models of interest.
- The attack is both sparse and effective.
***Weaknesses***
- The main issue with this paper is the lack of an intuitive explanation as to why this attack is better at finding sparse and effective adversarial examples than previous work. I would have liked to see a more detailed algorithmic comparison with previous work.


**Summary Of The Paper:**

This paper proposes the use of an evolutionary algorithm to construct decision-based black-box adversarial examples with L0 or sparsity constraints against image classifiers such as CNNs and Image Transformers. The algorithm uses an L2 distance constraint to check the fitness of a solution, and employs several tricks such as differential recombination and mutation to improve the quality of the solution. The experimental results demonstrate that the attack is more effective than the current SOTA sparse attacks, and is almost as effective as white-box attacks given enough queries.

**Summary Of The Review:**

Overall this is a solid paper that makes a reasonable contribution to a problem of some interest to the community.

+++++++++++++++++++++++++++++++++
Having read the rebuttal, I retain my score.

---

> ### Author Response · Authors · 2021-11-19
> **Algorithmic comparison with previous work**
>
> Thank you for your valuable feedback and your appreciation of our efforts.
>
> Our attack algorithm is better at finding sparse adversarial examples, because:
>
> 1) __Greedy vs. Evolutionary approach__. Pointwise chooses to greedily minimize the number of perturbed pixels by __randomly__ selecting and altering one dimension (i.e. single colour channel) of a randomly selected pixel position i,j  of the starting image $x' \in R^{C\times W \times H}$ at a time (i.e per query). If the alteration successfully fools the model, it will be retained; otherwise, the change will be discarded. In contrast, SparseEvo evaluates candidate proposals to alter several pixels at a time and all dimensions of a pixel simultaneously to yield new candidates solutions or offsprings for the next evolution; so it is able to converge faster and with less queries.
> 2) __Smaller search space__. Pointwise formulation leads to a search space with a size of $C\times W \times H$  where C is the three RGB channels, W is image width and H is image height. We reduce this search space to $W \times H$  because SparseEvo solely searches for pixel positions but does not try to search for different colors for each pixel (please kindly see “Defining a Dimensionality Reduced Search Space” in Section 3.2 and our response to __Reviewer 8jjY__).
>
> 3) __Better scalability to large image sizes__. Given that PointWise only changes one dimension at a time (i.e a pixel), to reduce the number of starting image (target class) pixel values different from the source image (to minimize L0), the random selection method needs to select: i) the same pixel position i,j; and ii) a different colour channel for the same pixel position i,j in subsequent iterations to move a given pixel value i,j in a starting image (target class image) to be the same as the source image. While this is more likely in a small image task (with smaller W and H values) like CIFA-10, it is far less likely, even within the 20,000 query budget used with large input images in the ImageNet task where mean sparsity values for the 1000 test image pairs remain nearly 1.
>
> 4) __Iterative improvements to “good” solutions__. Importantly, our approach formulates a search for a solution with the _minimum number_ of perturbed pixels through an iterative process of improving upon good solutions from previous iterations informed by our objective function. In contrast, Pointwise employs a purely random method to select the pixel dimension and position i,j to alter.
>
> Thus, SparseEvo is capable of searching for a desirable solution (an adversarial example with a smaller number of perturbed pixels) with much fewer queries.
>
> __Action (please see Rebuttal Revision)__
>
> - We added a __*new*__ discussion to the revised manuscript (see Appendix A.6)

---

### Public Comment · ~Maksym_Andriushchenko1 · 2022-02-02
**Comparison to Sparse-RS**

Dear authors,

We think our paper can serve as a strong baseline for your method:
[Sparse-RS: a versatile framework for query-efficient sparse black-box adversarial attacks](https://arxiv.org/abs/2006.12834) (on arXiv since June 2020)
as it is also a sparse attack which is black-box and aims at query efficiency.

In particular, Sparse-RS substantially outperforms the white-box PGD$_0$ in terms of the success rate (see Table 1 in our paper) which you use *"as an ideal case baseline"*.


Best,

Maksym

---

> ### Public Comment · ~Viet_Vo1 · 2022-02-14
> **Response to Maksym's suggestion**
>
> Thank you Maksym for your suggestion.
>
> The Sparse-RS attack assumes the adversary has access to the confidence scores from the victim model (a score-based threat model).
>
> But, our attack (SpaEvo) is designed for decision-based scenarios where solely the predicted label (hard label) is accessible to the adversary. In other words, SpaEvo and Sparse-RS are designed for attacking a victim model in different settings so they are not directly comparable.
>
> Nevertheless, it is interesting to study sparse attacks in score-based settings and it would be interesting to see how we can adapt our attack to a score-based setting to compare with Sparse-RS. So it could be our next research direction.

---

### Decision · Program_Chairs · 2022-01-20

**Decision:**

Accept (Poster)

**Comment:**

This paper introduces a technique to generate L0 adversarial examples in
a black-box manner. The reviews are largely positive, with the reviewers
especially commenting on the paper being well written and clearly explaining
the method. The main drawbacks raised by the reviewers is that the method
is not clearly compared to some prior work, but in the rebuttal the authors
provide many of these numbers. On the whole this is a useful and interesting
attack that would be worth accepting.